# Recent Advancements in Metallic Au- and Ag-Based Chitosan Nanocomposite Derivatives for Enhanced Anticancer Drug Delivery

**DOI:** 10.3390/molecules29102393

**Published:** 2024-05-19

**Authors:** Mahmoud A. El-Meligy, Eman M. Abd El-Monaem, Abdelazeem S. Eltaweil, Mohamed S. Mohy-Eldin, Zyta M. Ziora, Abolfazl Heydari, Ahmed M. Omer

**Affiliations:** 1Polymer Institute of the Slovak Academy of Sciences, Dúbravská Cesta 9, 845 41 Bratislava, Slovakia; abolfazl.heydari@savba.sk; 2Genomic Signature Cancer Center, Global Teaching Hospital, University of Tanta, Tanta 31527, Egypt; 3Chemistry Department, Faculty of Science, Alexandria University, Alexandria 21321, Egypt; emanabdelmonaem5925@yahoo.com (E.M.A.E.-M.); abdelazeemeltaweil@alexu.edu.eg (A.S.E.); 4Department of Engineering, Faculty of Engineering and Technology, University of Technology and Applied Sciences, Ibra 400, Oman; 5Polymer Materials Research Department, Advanced Technology and New Materials Research Institute (ATNMRI), City of Scientific Research and Technological Applications (SRTA-City), New Borg El-Arab City, P.O. Box 21934, Alexandria, Egypt; mmohyeldin@srtacity.sci.eg; 6The Institute for Molecular Bioscience, The University of Queensland, St Lucia, Brisbane, QLD 4072, Australia; z.ziora@uq.edu.au

**Keywords:** silver, gold, nanocomposite, drug delivery, cancer therapy, chitosan, cytotoxicity

## Abstract

The rapid advancements in nanotechnology in the field of nanomedicine have the potential to significantly enhance therapeutic strategies for cancer treatment. There is considerable promise for enhancing the efficacy of cancer therapy through the manufacture of innovative nanocomposite materials. Metallic nanoparticles have been found to enhance the release of anticancer medications that are loaded onto them, resulting in a sustained release, hence reducing the dosage required for drug administration and preventing their buildup in healthy cells. The combination of nanotechnology with biocompatible materials offers new prospects for the development of advanced therapies that exhibit enhanced selectivity, reduced adverse effects, and improved patient outcomes. Chitosan (CS), a polysaccharide possessing distinct physicochemical properties, exhibits favorable attributes for controlled drug delivery due to its biocompatibility and biodegradability. Chitosan nanocomposites exhibit heightened stability, improved biocompatibility, and prolonged release characteristics for anticancer medicines. The incorporation of gold (Au) nanoparticles into the chitosan nanocomposite results in the manifestation of photothermal characteristics, whereas the inclusion of silver (Ag) nanoparticles boosts the antibacterial capabilities of the synthesized nanocomposite. The objective of this review is to investigate the recent progress in the utilization of Ag and Au nanoparticles, or a combination thereof, within a chitosan matrix or its modified derivatives for the purpose of anticancer drug delivery. The research findings for the potential of a chitosan nanocomposite to deliver various anticancer drugs, such as doxorubicin, 5-Fluroacil, curcumin, paclitaxel, and 6-mercaptopurine, were investigated. Moreover, various modifications carried out on the chitosan matrix phase and the nanocomposite surfaces to enhance targeting selectivity, loading efficiency, and pH sensitivity were highlighted. In addition, challenges and perspectives that could motivate further research related to the applications of chitosan nanocomposites in cancer therapy were summarized.

## 1. Introduction

Cancer has emerged as a significant disorder responsible for a substantial number of fatalities annually on a global scale [1]. The National Cancer Institute reported that, as of January 2022, an estimated 18.1 million cancer survivors resided in the United States, comprising approximately 5.4% of the population. Projections suggest that this number will increase by 24.4% to 22.5 million by 2032 and is anticipated to rise to 26.0 million by 2040 (Figure 1A). Figure 1B shows the distribution of cancer survivors in the U.S. by age as of January 1, 2022. The majority of survivors are 65 years and older, followed by those aged 40–64 years. The age groups of 20–39 years and 0–19 years have smaller portions of survivors. Each year, a significant number of people die from cancer, necessitating a continuous requirement and desire for the development of potent drugs to treat diverse types of malignancies [2]. Cancer is characterized by the abnormal proliferation and growth of many cells inside the body [3]. Within the human body, cellular replacement occurs in response to cellular death, damage, or aging. Cell division is a fundamental biological process that facilitates the generation of new cells within the human body. It involves the multiplication and expansion of normal, healthy cells. The disruption of this system can lead to the development and multiplication of aberrant cells, specifically malignant cells [4]. Moreover, the process of destabilizing, degrading, and invading healthy tissues in cancer disrupts the ongoing growth of cells [5].

The development of cancer can be attributed to various factors, including virus infections, smoking, inadequate dietary habits, alcohol consumption, obesity, exposure to ionizing radiation, and hereditary predisposition [6]. The treatment must be appropriately determined for each specific situation, considering the type of cancer. The increase in cancer rates is often attributed to the environmental and climatic changes resulting from industrialization, as well as alterations in lifestyle and dietary patterns [7]. Therefore, if the symptoms indicate the early stage of the disease, the inclusion of previous diagnostic evaluations will have a minimal impact on improving cancer outcomes [8].

The delivery of anticancer drugs faces numerous challenges and difficulties, including the abundance of drugs and their corresponding physicochemical characteristics, the toxicity of administration, the potential difficulty of targeting a specific tumor location, and the complexity of treatment plans. To provide more clarification, the challenges associated with the supply of anticancer drugs can be categorized based on the administrative route [9]. The oral delivery route is a straightforward and preferable method for treating various gastrointestinal tumors, such as colorectal cancer, stomach cancer, pancreatic cancer, and anal cancer [10]. Each type of cancer may require different treatment approaches depending on its location, stage, and other factors [11]. Therefore, the drugs can be absorbed directly through the gastrointestinal tract, where many of these tumors are located, potentially leading to higher drug concentrations at the tumor site. This targeted delivery approach enhances the therapeutic effect while minimizing systemic toxicity [12]. However, the oral administration of anticancer medications presents several obstacles, including the need to improve bioavailability (solubility and/or permeability), reduce enzymatic degradation, and achieve targeted delivery to specific sites within the gastrointestinal tract [13,14]. Novel oral formulations containing non-cytotoxic targeted agents [15], lipid-based formulations [16], permeation enhancers, and gastro-retentive dosage forms [16,17] have the potential to address these challenges. The most prevalent method of delivering cancer medications is through intravenous injection, which promises a high level of absorption and minimal instability between and within patients. Several challenges arise with this delivery method, such as the unfavorable pharmacokinetic parameters of the drug, the non-specificity of some chemotherapeutic agents, leading to severe side effects (off- and on-target toxicity), and the difficulty in traversing highly impermeable barriers, such as the blood–brain barrier [9]. In addition, the subcutaneous route is frequently associated with reducing local toxicity, facilitating the injection of highly concentrated forms, providing consistent doses over an extended duration and controlling numerous release rates [9].

Nanomedicines and nanocarriers represent encouraging approaches for delivering anticancer medications with improved accuracy and effectiveness [18,19]. Nanomedicines involve the encapsulation or attachment of therapeutic substances to nanoparticles, while nanocarriers are specialized vehicles crafted to transport drugs to precise locations within the body [20,21]. In the realm of anticancer treatment delivery, both nanomedicines and nanocarriers present multiple benefits [22]. Their nanoscale size enables them to traverse biological barriers more efficiently, including tumor tissues with impaired blood vessel function. This enhanced permeability and retention (EPR) effect enables preferential accumulation of the drug at the tumor site, minimizing exposure to healthy tissues and reducing systemic toxicity [23]. Furthermore, nanocarriers have the capability to be tailored to target specific molecular markers that are excessively expressed on cancer cells, thereby refining the precision of drug delivery. Moreover, nanocarriers can protect encapsulated drugs from degradation and premature clearance in the bloodstream, prolonging circulation time and enhancing bioavailability. This feature of controlled release enables sustained delivery of drugs, maintaining therapeutic levels over an extended duration and potentially decreasing the frequency of administration [24]. Additionally, the adaptability of nanomedicines and nanocarriers permits the simultaneous delivery of multiple drugs or therapeutic agents with complementary modes of action. This coordinated approach can bolster treatment effectiveness, surmount drug resistance, and diminish the likelihood of tumor recurrence [25]. The primary novel methodologies now rely on the utilization of nanomedicines and nanocarriers and ought to be expanded for the transportation of biomolecules [26,27].

The utilization of nanotechnology in anticancer drug delivery presents several challenges, including the encapsulation of hydrophilic or hydrophobic pharmaceuticals, regulation of drug release, protecting the loaded drugs from degradation, and improving drug absorption and penetration [28]. Nevertheless, there are certain limitations for using nanotechnology, including poor drug loading, rapid or burst release before reaching the intended target, and probability of clearance through renal filtration owing to nanoscale dimensions. The utilization of polymer–prodrug systems and the integration of polymer nanocomposites can overcome these drawbacks and enable the creation of a depot beneath the skin, facilitating the controlled and gradual release of drugs with minimal adverse effects on the surrounding region [29,30].

Drug-encapsulated nanocomposites offer numerous benefits, such as enhanced pharmacokinetics and the ability to selectively administer medications to specific locations or tumors [31].

Metallic nanoparticles, including gold nanoparticles (AuNPs) and silver nanoparticles (AgNPs), are utilized in clinical applications. This is due to their unique forms, sizes, and surface-dependent properties [32]. Gold nanoparticles (AuNPs) are highly effective in the field of cancer diagnostics. The incorporation of anticancer drugs into colloidal AuNPs has the potential to overcome drug resistance, resulting in a reduction in the required dosage and, thereby, modifying the adverse effects on healthy cells [33]. AgNPs have demonstrated favorable conductivity and chemical stability [34]. Furthermore, they have been investigated as potential carriers for delivering therapeutic substances to diseased cells. Nevertheless, AgNPs tend to aggregate and form bigger clusters that diverge from the nanoscale. This reduces the efficacy of a nano-delivery system and poses challenges to its practical implementation [35]. Hence, it is necessary to incorporate further organic or biological surface coatings to enhance the stability of AuNPs and AgNPs [36,37]. Thus, it is widely acknowledged that natural polysaccharides and natural biopolymers are advantageous stabilizers. Therefore, metal nanoparticles tend to be aggregated in biological fluids, and this can be overcome by the generated electrostatic attractive forces between polysaccharide (such as amino groups of chitosan) and metallic nanoparticle [38]. Consequently, this can provide an effective driving force for the formation and stabilization of the Au and AgNPs [39,40].

The chitosan (CS) biopolymer is a derivative of chitin that has undergone deacetylation, resulting in notable biological characteristics [41,42]. These characteristics include reduced toxicity, enhanced biocompatibility, stability, and effective adhesion to mucus membranes. The utilization of chitosan in drug administration facilitates the pH selectivity of nanometals, as chitosan is a cationic biopolymer that is suitable for drug delivery purposes [43]. CS molecules possess NH_2_ and OH functional groups that serve as chelating sites for drugs within target cells, depending on their pH sensitivity. In addition, CS demonstrates the ability to selectively target specific sites for drug delivery, resulting in an enhanced therapeutic index of the drug in comparison to alternative natural materials [44,45]. Additionally, it exhibits antibacterial characteristics and improves the use of chitosan in drug encapsulation. Nevertheless, CS exhibits some limitations, such as reduced mechanical strength, limited solubility in both neutral and alkaline pH environments, and challenges associated with pore size adjustment [46,47].

In the current review, we highlighted the role of nanotechnology in cancer treatment in addition to nanocomposites and their applications for anticancer drug delivery. Moreover, the recent advances in chitosan and its nanocomposites with bioactive metallic nanoparticles (Ag and Au) for the effectual delivery of various anticancer drugs were investigated.

## 2. Nanotechnology for Cancer Treatment

Nanotechnology is the scientific study of materials that exist at a scale of 10^−9^ m [48]. Nanotechnology is widely regarded as a means of scientific progress, as demonstrated by the significant increase in commercial products, academic publications, and patents in the medical, pharmaceutical, and cosmetic sectors. Pharmaceutical and biotechnology companies are interested in improving medications that exhibit limited efficacy due to factors, such as poor solubility, toxicity, aggregation, poor mobility, rapid degradation in living organisms, or a short half-life [49,50]. Nanoparticles (NPs) have garnered attention as carriers for anticancer medications because of their advantageous properties, such as enhanced permeability and retention effect. In addition to diminishing the negative impacts of the medication, nanoparticles (NPs) have been found to provide protection against drug degradation and enhance the drug’s bioavailability [51,52].

There are still difficulties in the fabrication of metal nanoparticles (NPs) for purposes, such as imaging, drug delivery, diagnosis, and treatment. The development of nanoparticles remains challenging due to the observed instability of several nanoformulations in biological fluids [51,53]. Biological fluids’ high ionic strengths often induce NP aggregation and losing their colloidal stability, negatively affecting their function. The high content of biomacromolecules, including lipids, sugars, nucleic acids, and proteins, also affects NPs’ stability and viability for various applications [54]. Conventional approaches of cancer treatment are associated with negative consequences, including limited solubility, insufficient bioavailability, frequent deterioration, lack of specificity, and drug resistance. In addition, the high doses of conventional medications cause adverse effects in specific anatomical places, such as the skin, hair, RBCs, bone, genitourinary system, and lymphatic system [6].

NPs possess the capability to transport pharmaceuticals directly to cancerous cells, hence enhancing their efficacy. Nevertheless, the usefulness of nanoparticles is contingent upon the growth stage and type of tumor. The utilization of nanoparticles (NPs) in the field of cancer diagnosis and treatment has seen significant advancements, enabling the identification and treatment of single cancer cells through the targeted delivery of carriers. NPs, such as carbon nanotubes (CNTs), calcium nanoparticles (CaNPs), graphene, and polymeric NPs (including chitosan), have demonstrated enhanced capabilities in cancer diagnosis and treatment owing to their significant dimensions, surface charge, and morphology. NPs undergo functionalization with various biological molecules, such as antibodies, thereby facilitating the administration of drugs and the detection of cancer cells [6,55].

### 2.1. Role of NPs in Cancer Treatment

Nanoparticles can enhance the concentration of medication within cancer cells and direct therapy toward solid tumors via either passive or active targeting mechanisms, as demonstrated in Figure 2 [56]. An active targeting mechanism involves the incorporation of targeting ligands or molecules onto the surface of nanocarriers, enabling selective interactions and binding to cellular receptors. The effectiveness of drug delivery depends on the overexpression or coating of nanoparticles by either cancer cells or angiogenic endothelial cells, which enhance the internalization of nanoparticles and improve their efficacy [57]. The differentiation between tumor cells and healthy cells is facilitated by the presence of distinct molecular markers within the different cell types comprising the tumor bulk. Moreover, active targeting exhibits a high degree of selectivity, remarkable versatility, and a reduced incidence of side effects.

The passive targeting mechanism plays a crucial role in anticancer targeting therapy. In this mechanism, non-targeted nanoparticles can enter tumor sites through the sieve-like microvasculature of solid tumors, facilitated by permeability factors such as bradykinin and nitric oxide. This phenomenon is commonly referred to as the retention (EPR) effect [58]. Drug-loaded nanocarriers tend to accumulate within the tumor microenvironment (TME) because of their considerable size, whereas smaller molecules are liable for back diffusion and efflux. However, the EPR effect is not sufficient for an effective accumulation of molecules with low molecular weight near the tumor site [59], which are less selective, restricted in use, and cause more side effects [31].

For applications involving cancer, it is crucial to regulate the factors, such as the shape, surface charge, homogeneity, stability, and toxicity, of nanoparticles. The physicochemical properties are also influenced by the production process. Various materials can be exploited in these processes, such as metals, lipids (liposomes, solid lipid nanoparticles), and polymers (nanoparticles, micelles, and dendrimers) [60].

### 2.2. Nanocomposite in Cancer Therapy

Nanocomposites are often formed through the dispersion of a matrix phase with fillers, such as nanoparticles, nanolayers, and nanotubes. However, the classification of nanocomposites can vary depending on the material composition of each phase, resulting in three distinct forms. The organic–organic nanocomposite comprises organic nanoparticles distributed inside an organic polymer or lipid matrix phase. Furthermore, inorganic–inorganic nanocomposites consist of a matrix and dispersed phases made of one or more minerals. The last type of composite material is referred to as organic–inorganic nanocomposites, comprising an organic polymer matrix phase and a nanometric form of an inorganic filler phase [60]. Similarly, the utilization of nanocomposites in cancer therapy needs the process of modifying nanoparticles with biomolecules and incorporating medications into them to enhance their efficacy in tumor interaction. The hydrophilic polymer matrix leads to enhanced solubility of the nanoparticle and greater compatibility. Figure 3 depicts a proposed schematic design that provides a clear illustration of the synthesis process of the nanocomposite, which extends the potential for application in drug delivery for cancer treatment.

Nanocomposites in drug delivery systems are used to avoid cancer cell treatment resistance. The literature documents the synthesis and manufacture of many conjugated materials, such as graphene–gold [61], chitosan–palladium [62], gold ferrite [63], and graphene oxide/bismuth selenide [64], among others, for drug delivery [65]. Furthermore, the presence of a magnetic field gradient facilitates the vectorization of nanocomposites within the tumor, enabling the administration of medication. Some examples of possible materials include nano-graphene [66], amine-polyglycerol functional shell-modified silica-coated magnetic iron oxide [67], and hybrid protein-inorganic nanoparticles [68]. The pH level within the tumor microenvironment poses a significant challenge that nanocomposites must address. Cancer exhibits accelerated proliferation compared to healthy cells, resulting in a deficiency of oxygen and nutrients, leading to acidosis and a decrease in pH. Researchers have investigated pH-sensitive nanocomposite systems, like superparamagnetic Fe_3_O_4_ nanoparticles and magnetic nanoparticles made from chitosan, to treat cancer more effectively [69,70]. Research findings indicate that nanocomposites possess the ability to breach biological barriers and deliver drugs to specific cells within the tumor microenvironment. This consequently enhances the effectiveness of cancer therapy [71].

## 3. Chitosan and Its Nanocomposites for Cancer Treatment

Chitosan (CS) is a biopolymer made up of β-(1-4)-linked glucosamine and N-acetyl-D-glucosamine units [72]. It is a linear, cationic polysaccharide produced from simple deacetylation of the chitin biopolymer, which is present in the exoskeleton of insects, marine aquatic animals, and microorganisms, like fungi, yeast, and microalgae [73]. Chitosan has drawn much attention in diverse medical and pharmaceutical fields, such as tissue engineering, wound healing, and drug delivery [74]. This is due to its excellent characteristics, including low-cost production, non-toxic, mucoadhesive, biocompatibility, biodegradability, low immunogenicity, and antimicrobial and anticancer activities [41,46]. Owing to its cationic nature, adequate stability, versatility in formulation, and ease of modification, CS exhibits considerable potential as a viable option for the formation of nanoparticles. Various methods have been employed to fabricate CS NPs, such as ionotropic gelation, spray drying, crosslinking via water-in-oil emulsion, reverse micelle formation, emulsion-droplet coalescence, nanoprecipitation, and self-assembling techniques [75,76].

The hydrophilic drug can permeate epithelial cells by adhering to negatively charged cell membranes and then opening the tight connection between them. This process extends the retention of the medication and enhances cellular uptake [77]. Hence, many chitosan derivatives, such as chitosan–thymine conjugate, carboxymethyl chitosan, sulfated chitosan, sulfated benzaldehyde chitosan, and polypyrrole–chitosan, have exhibited encouraging anticancer properties against a wide range of cancer cells [78]. Chitosan derivatives can enhance their anticancer properties by combining them with nanocomposites or other chemical agents. The bioavailability of chitosan in tumor cells was improved through the conjugation of 3-amino-2-phenyl-4(3H)-quinazolinone on a nanocomposite of PPC-silver chloride. The formulation described in reference [68] effectively captured molecules from non-cancerous cells and demonstrated a prolonged release of chitosan into cancer cells. Figure 4 shows different formulations of CS-based multifunctional targeted nanosystems for the simultaneous imaging and therapy of cancer [56].

### 3.1. Chitosan-Au Bio-Nanocomposite for Anticancer Drug Delivery

The utilization of AuNPs for drug administration is of extreme importance owing to their claimed biocompatibility and non-cytotoxicity [79,80,81]. Additionally, metal nanoparticles exhibit lower toxicity compared to other substances. They have a high surface-to-volume ratio, allowing them to encapsulate pharmaceuticals and serve as carriers for drug delivery [82]. Moreover, they exhibit exceptional infiltration into cancer tissues and, notably, lack immunogenicity within the human body. Additionally, this material exhibits distinct physical and chemical properties, including the influence of nanoscale, surface, quantum, electrical, and optical phenomena, hence extending specific advantages. In the field of photothermal therapy (PTT), gold nanoparticles (AuNPs) have been employed as a nanoscale-based photothermal agent to elevate the temperature of specific tissues through the conversion of light into heat [83]. This phenomenon can be attributed to their enhanced light absorption capabilities and increased stability under exposure to light. PTT specifically eliminates cancer cells, which are more exposed to an increase in temperature. AuNPs have certain limitations that hinder their application in the pharmaceutical industry. The toxicity of uncoated gold nanoparticles (AuNPs) is dependent upon various factors, including their composition, shape, size, coating, charge, hydrophobicity, solubility, reactivity, and the nature of biological medium qualities. Research has demonstrated that naked AuNPs tend to accumulate within lysosomes, resulting in alterations to their optical characteristics and how they are activated by radiation. This review discusses the application of chitosan-dispersed gold nanoparticle composites to load various anticancer medications, addressing the challenges associated with their encapsulation and mitigating the cytotoxic side effects. Table 1 presents the advantages of modifying matrix chitosan and dispersed gold nanoparticles to load various anticancer drugs.

#### 3.1.1. Chitosan-Au Nanocomposite for Doxorubicin Delivery

Alle Madhusudhan et al. [84] prepared a composite of chitosan AuNPs that may load doxorubicin (DOX) according to the scheme represented in Figure 5 and exhibit anticancer properties. This composite is capable of targeting cancer cells under intracellular acidic conditions. The chitosan matrix phase was modified to carboxymethyl chitosan (CM-CS) to serve as a capping and reducing agent for the synthesis of AuNPs. This modification demonstrated the remarkable stability of AuNPs at different electrolyte concentrations and pH values. The ionization of carboxylic groups occurred at an acidic pH, resulting in increased drug release and enhanced pharmacological activity at an acidic pH that is suitable for targeting cancer cells. The structure of CM-CS is like amino acids, as it contains both amino and carboxylic groups. This unique composition enables effective drug interaction, particularly for DOX-like drugs. These drugs contain several amines and hydroxyl functional groups, which possess the ability to form hydrogen bonds with CM-chitosan-activated AuNPs. The covalent connections on the surface of AuNPs are strengthened by the hydrogen bonding and electrostatic attraction between the carboxylic groups of CM-CS and the amino groups of DOX molecules. Furthermore, cervical cancer cells exhibited a higher uptake of DOX when loaded with AuNPs compared to free DOX, suggesting a potential solution to address the issue of drug resistance in cancer. The cytotoxicity of free DOX is enhanced by CMC-AS-dispersed AuNP-loaded DOX.

#### 3.1.2. Chitosan-Au Nanocomposite for 5-Fluroacil Delivery

Nivethaa et al. [85] fabricated bimetallic nanoparticles through the combination of gold nanoparticles with other metal nanoparticles. These nanoparticles were then dispersed into chitosan composites to facilitate the loading of 5-Fluroacil, an anticancer drug. Silver nanoparticles were incorporated into AuNPs, hence increasing their antibacterial efficacy. This approach aims to integrate the unique properties of Au (photothermal) and Ag nanoparticles within nanoparticle alloys comprising both metals. This merger aims to expose novel paths for research and medical treatment, owing to the distinctive properties exhibited by this alloy, which are significantly different from those of single metal components. The study involved the in situ reduction of AgNO_3_ and HAuCl_4_ in a CS solution, resulting in the formation of Ag-Au nanoparticles. A CS/Ag-Au fluorescent nanocomposite was prepared using a simple and cost-effective chemical process, with varying quantities of bimetallic nanoparticles distributed in chitosan. The manufactured composite has a distinct emission peak within the near-infrared range (700–900 nm), making it suitable for cellular imaging applications. The process of loading 5-FU drugs results in duplication of nanoparticle size, leading to a loading capacity of 97%. The incorporation of bimetallic nanoparticles resulted in an accelerated release of 5-FU, reaching 67.6% after 72 h, in contrast to the 97% release rate observed with a single metal dispersion. The cytotoxicity assessment of the nanocomposite containing 5-FU against MCF-7 cancer cells demonstrated that the bimetallic nanoparticles combined with the chitosan composite exhibited more efficacy compared to the individual Ag, Au, and CS particles.

#### 3.1.3. Chitosan-Au Nanocomposite for Curcumin Delivery

Nadia G. Kandil et al. [86] developed another photothermal chitosan–gold nanoparticle composite to load curcumin, which can be used in cancer treatments. To incorporate a recently modified curcumin derivative, metal oxide (ZnO) nanoparticles and gold (Au) nanoparticles were combined within a chitosan mixture. This modified the mechanism by which curcumin specifically targeted cancer cells. The researchers prepared a new hydrazinocurcumin derivative (HCUR) by a thermal condensation reaction of curcumin, a polyphenol that is found naturally in the rhizome with a pyridazine derivative (VII). Pyridazines are an important group of heterocycles that have gotten a lot of attention in science, especially in the pharmaceutical field, because they have a wide range of biological activities and anticancer activity.

After that, the ZnO and Au nanoparticles that were dispersed in chitosan were used to load the HCUR composite that had been made using a self-assembling mechanism. The entrapment efficiency (EE%) exhibited a range of 39.41% to 73.70%. The Au-HCUR NPs showed the highest percentage (73.70%) but less than the capacity for loading 5-Fluoracil using a bimetallic silver and gold nanoparticle composite. A prolonged drug was released, and it was noticed that it happened faster at an acidic pH than at a neutral pH. This is because the nanoparticles diffuse better in the acidic environment. The study found that the type of cell had an impact on cell viability, indicating that NP formulations were more effective against HCT-116 cells (colon cancer) than CUR. Notably, CS-HCUR NPs show the highest level of activity in terms of cell viability. Also, ZnO-HCUR nanoparticles had the strongest effect on destroying HepG-2 cells, which are a type of hepatocellular carcinoma. According to this study, nanoformulations are very suitable for medicinal use as anticancer agents.

#### 3.1.4. Chitosan–Au Nanocomposite for Paclitaxel (PTX) Delivery

Paclitaxel (PTX) has demonstrated efficacy in the treatment of breast and ovarian cancer; however, its solubility is limited, and its therapeutic use is limited [89,90]. Manivasagan et al. prepared a nanocomposite of chitosan AuNP-loaded PTX [87] to overcome the limitations of PTX as a drug carrier, ensuring minimal side effects. These composites were intended for use as contrast agents in photoacoustic imaging (PAI) and photoacoustic tomography (PAT). The utilization of contrast agents such as nanoparticles and the application of photoacoustic imaging (PAI), a more recent technological advancement, enable the imaging of tissues and cells. A modified chitosan matrix phase was used to disperse gold nanoparticles (AuNPs) that can work with biological systems. This matrix phase consisted of chitosan oligosaccharide (COS) as a reducing and stabilizing agent. Subsequently, paclitaxel (PTX) was loaded onto the AuNPs. The study demonstrated that PTX-COS AuNPs exhibited sustained and pH-dependent drug release profiles. Additionally, these nanoparticles were recognized as a unique class of optical contrast agents for photoacoustic medical imaging. It is worth noting that no prior research has explored the utilization of PTX-COS AuNPs as innovative agents for drug delivery. The PTX-COS AuNPs have shown effective cytotoxic properties against MDA-MB-231 cells by inducing apoptosis, resulting in increased formation of reactive oxygen species (ROS) and changed levels of matrix metalloproteinases (MMPs). Additionally, these AuNPs exhibited remarkable cellular uptake in MDA-MB-231 cells. In conclusion, PTX-COS AuNPs exhibit ability as advanced anticancer agents for the treatment of breast cancer and have potential as a novel contrast agent for photoacoustic imaging (PAI). By adding new gold nanoparticles (AuNPs) to this composite material, their unique physical and chemical properties, like surface plasmon resonance (SPR), as well as their optical properties, enhance the composite’s capability to permit surface modification for biomedical applications [91]. In addition, green synthesis, a cost-effective, zero-energy, and less time-consuming approach to synthesizing AuNPs without the utilization of toxic chemicals, has been documented [92]. However, the main challenge of CS is its poor solubility in neutral pH conditions. Chitosan oligosaccharide (COS) has garnered significant interest in the field of biomedical applications due to its ability to overcome these challenges and its potential as a highly suitable option for drug administration [93].

#### 3.1.5. Chitosan-AuNPs for 6-Mercaptopurine (6MP) Delivery

A composite of chitosan-dispersed gold nanoparticles (AuNPs) was synthesized to load 6-mercaptopurine (6MP), a therapeutic agent employed in the treatment of cancerous solid tumors [94]. Faid et al. [88] developed chitosan-encapsulated 6MP with subsequent loading on AuNPs to form 6MP-CNPs-AuNPs nanocomposites, as depicted in Figure 6. The formulated nanocomposite has the advantage of reducing the side effects of 6MP, like other chemotherapeutic agents, such as allergic reactions and hepatotoxicity [95]. The researchers proposed encapsulating 6MP into chitosan nanoparticles (CNPs) to form chemically linked 6MP-CNP complexes. These complexes additionally loaded onto AuNPs, resulting in the development of a novel nanocomposite consisting of 6MP-CNPs-AuNPs. This nanocomposite facilitates therapeutic effects for chemo-photothermal techniques. Chitosan nanoparticles (CNPs) are prepared using an ionic gelation method. To make the nanoparticles, a tri-polyphosphate (TPP) polyanionic crosslinker and positively charged chitosan strands combine electrostatically [96]. In addition, the 6MP-CNP composite was prepared using the modified ionic gelation process. This composite was then loaded to AuNPs that were synthesized using the traditional wet chemical method, with tri-sodium citrate serving as a capping and reducing agent. Chitosan nanoparticles (CNPs) improve the targeting of drugs to cells by absorbing the loaded or encapsulated drugs [97]. The composite fabricated from CNPs and gold nanoparticles demonstrates exceptional stability in zeta potential. The encapsulation efficiency of 6MP was determined to be 57%. The cytotoxicity of 6MP-CNPs and 6MP-CNPs-AuNPs on the MCF7 breast cell line was significantly increased, resulting in IC50 values of 9.3 and 8.7 m, respectively. The combined effects of nanocomposites were more effective at treating cancer than the effects of chemotherapy and photothermal therapy used alone. A decrease in the IC50 values to 5 and 4.4 m, respectively, indicated a greater reduction in cell viability following the application of a diode laser to 6MP-CNPs and 6MP-CNPs-AuNPs.

### 3.2. Chitosan-Ag Nanocomposite

Silver nanoparticles (AgNPs) are frequently used in composites for biomedical purposes to increase the surface area for drug encapsulation and improve the antibacterial efficacy [98]. When chitosan-dispersed silver nanoparticle composites are used as natural carriers for different anticancer drugs, they reduce the adverse side effects and improve the treatment profile. Ways to reach this approach through different modifications to challenge the properties of the composite are summarized in Table 2. Before that, different anticancer drug-loaded chitosan Ag nanocomposites were fabricated and explained as follows.

#### 3.2.1. Chitosan-Ag Nanocomposite for 5-FU Delivery

Different modification processes have been performed on chitosan-dispersed AgNPs for improving the loading of 5-FU anticancer drugs despite the following challenges.

##### Modification of Chitosan Matrix Phase

The challenge to prepare a pH-sensitive controlled intestinal discharge of anticancer 5-FU medicine was approached by Hanna et al. [99], according to the schematic diagram in Figure 7. Silver nanoparticles (SNPs) were synthesized and dispersed within chitosan composite carriers to deliver them orally to the intestines composed of a polyelectrolyte complex (PEC)-modified chitosan matrix phase. The matrix phase of the PEC consists of chitosan that has been modified with N, N, and N trimethyl groups to enhance its solubility in water across a wide pH range (pH 2–9) and carboxymethyl Kappa-carrageenan that is derived from chloroacetic acid and Kappa-carrageenan. It is one of the carrageenan derivatives that is commonly employed in biomedical applications [105]. Silver nanoparticles were dispersed in various ratios using a one-pot green biosynthesis method. The encapsulation efficiency was further enhanced by increasing the amount of AgNPs, resulting in a peak efficiency of 92% with the maximum nanoparticle content. In addition, the incorporation of silver nanoparticles (AgNPs) leads to the prolonged release of 5-FU, achieving a 96% release rate within 24 h at a pH of 7.4. This release is attributed to distinct kinetic mechanisms at a pH of 1.2. The treated HCT116 cells exhibited significant cytotoxicity, as evidenced by an IC50 value of 31.15 µg/mL and an apoptotic cell count of 30.66%.

##### Dispersed-Phase AgNPs Modification with CNT

In a study by Nivethaa et al. [100], a chitosan-incorporated sliver nanoparticle-encapsulated 5-Fu anticancer drug composite was prepared using chemical methods. The researchers observed release and cytotoxicity results that were like those reported in a previous study by Hanna et al. [99]. This study aimed to prepare systems with more prolonged release time to extend the dosage intervals. The incorporation of carbon nanotubes (CNTs) into the composite was preferred due to the numerous advantages they offer for biomedical applications. Hence, the inclusion of carbon nanotubes (CNTs) prolongs the sustained release of 5-FU to 63.1% after 72 h, compared to 88.1% for the CS-Ag nanocomposite without CNT. This is attributed to the relatively strong binding of 5-FU to multi-walled carbon nanotubes (MWCNTs). Furthermore, the IC50 value serves as an indicator of the comparative effectiveness of the CS/Ag nanocomposite encapsulated with 5-FU in destroying carcinogenic MCF-7 cells, in addition to its cytotoxic properties. This can be attributed to the characteristic capability of carbon nanotubes to facilitate targeted drug delivery to cells and tissues because of their ability to penetrate the cellular membrane and enter cellular components without inducing observable harm to the cells [100].

##### Dispersed-Phase AgNP Modification with GO

Another enhancement for the chitosan composite is dispersed silver nanoparticles to encapsulate 5-FU shifted towards emerging photothermal properties, along with antimicrobial and release sensitivity, which enhance cytotoxicity against tumor cells by adding two different components.

Su et al. [101] reported the incorporation of graphene oxide (GO) into the chitosan composite. Because the composite was formulated with unique physiochemical properties, it could be used in many biomedical areas, such as drug delivery, photothermal therapy, and tissue regeneration [106,107]. The unique properties of each component material were fully utilized for the first time in this kind of composite system. These include the antimicrobial properties of dispersed AgNPs that are made using green or chemical methods, the pH responsiveness of CS, and the photothermal conversion property of GO. The new multiple functions that the prepared composite system showed could make it suitable for many applications, such as antimicrobial, releasing fluorouracil drugs, and photothermal cancer treatment.

#### 3.2.2. Chitosan-Ag Nanocomposite for Dox Delivery

Rasoulzadehzali et al. [102] prepared a bio-nanocomposite consisting of chitosan AgNPs for the controlled release of Dox. The fabrication of the bio-nanocomposite involved the dispersion of graphene oxide (GO) onto silver (Ag) nanohybrid particles. This process resulted in the formation of unique pH-sensitive bio-nanocomposite hydrogel beads, which were crosslinked using sodium, a polyanion crosslinking agent, tripolyphosphate (STPP) [108]. The dispersion of GO-Ag nanohybrid particles within the chitosan composite exhibits a notable enhancement in drug loading capacity, as well as prolonged drug release. In addition, it was observed that the release percentage of Dox was higher at pH 1.2 compared to pH 6.8. This can be attributed to the protonation of the NH_2_ groups of chitosan under acidic conditions, resulting in increased swelling and subsequently leading to higher release performance. The bio-nanocomposite that was prepared did not exhibit any notable toxicity on SW480 cells when DOX was not loaded. However, the burst release behavior of the loaded bio-nanocomposite led to a considerable reduction in the viability of these cells. This characteristic makes it a promising candidate for cancer therapy.

#### 3.2.3. Chitosan-Ag Nanocomposite for Curcumin Delivery

Curcumin (CUR), an anticarcinogenic agent [109], was selected as a representative of extremely water-insoluble hydrophobic drugs that exhibit limited bioavailability, significant instability in aqueous basic environments, and a higher rate of degradation in the gastrointestinal tract (GIT). These factors pose challenges to the conventional oral administration of CUR and restrict its ability to be loaded onto carriers [110]. Therefore, there has been increased attention towards preparing novel-controlled drug delivery systems (DDS) for gastrointestinal (GIT) purposes. These systems aim to improve the effectiveness of targeting the intestines while minimizing stomach discharge and reducing the overall toxicity to the body. Therefore, El-Maadawy et al. [103] successfully prepared a composite material consisting of chitosan-dispersed AgNPs to encapsulate CUR. The composite matrix phase was modified to form a polyelectrolyte complex (PEC) that serves as a pH-sensitive carrier for the efficient and controlled delivery of CUR to cancer cells through the gastrointestinal tract (GIT). This approach is similar to the modified chitosan composite previously fabricated by Hanna et al. [99], which was designed to load 5-FU and exhibit pH sensitivity in the intestine. The chitosan matrix phase has been modified with N, N, N-trimethyl chloride formation (TMC), which exhibits water solubility through a wide pH range of 2 to 9. This modified matrix phase is also utilized for reducing silver-forming nanoparticles to be dispersed, with the inclusion of carrageenan. This incorporation of carrageenan is increasingly recognized as a biomedical benefit for the composite material. The drug-loaded carrier, AgNPs, exhibited significant antibacterial efficacy against various bacterial strains. Furthermore, the novel carrier (CUR-loaded AgNPs) exhibited a non-cytotoxic effect on human normal cells, demonstrating favorable biocompatibility. However, it also induced apoptosis in malignant cells, resulting in a significant cytotoxic effect. Therefore, this is proposed to be a safe oral formulation in the controlled transport of hydrophobic drugs. The release capacity of curcumin, while maintaining its bioactivity, was found to be greater in simulated intestinal fluid (SIF, pH 7.4) compared to gastric fluid (SGF, pH 1.2), with a sustained efficacy of up to 24 h. In addition, it was observed that the loading capacity and encapsulation efficiency of CUR increased with an increased dispersion of silver nanoparticles in an acidic solution. This may be due to the properties embedded in composite matrix phase modification based on chitosan modification and the incorporation of carrageenan. Furthermore, the adsorption of CUR on the nanoparticles facilitates an improvement in the drug loading capacity [111], thereby promoting the controlled release of the loaded drug [112].

#### 3.2.4. Chitosan-Ag Nanocomposite for Cisplatin (CIS) Delivery

CIS, like most anticancer drugs, lacks sensitivity and specificity, leading to unpleasant side effects in patients because of administering high therapeutic doses. Hence, there is a need for novel treatment strategies to mitigate these problems. Although limited studies have been conducted on the evaluation of CIS-conjugated AgNPs functionalized with a CS biopolymer, S. Gounden et al. [104] evaluated the sensitivity and specificity of CS-functionalized AgNPs as delivery vehicles of CIS to human cancer cells. They synthesized AgNPs, functionalized with chitosan (CS), and loaded with the anticancer drug cisplatin (CIS) to be studied against Caco-2 (colorectal adenocarcinoma), MCF-7 (breast adenocarcinoma), HepG2 (hepatocellular carcinoma) and SKBR-3 (breast adenocarcinoma), and the non-cancer HEK293 (embryonic kidney) cells. The delivery system of AgNPs functionalized with chitosan exhibited desirable properties, including small size, a high positive zeta potential, and the ability to encapsulate over 80% of cisplatin. This cisplatin was released rapidly from the nano-complex at low pH, making it suitable for delivery to the tumor microenvironment. The compound demonstrated a high degree of selectivity towards breast cancer cells, resulting in more than 80% cell death at dosages below 10 μg. Additionally, it exhibited no cytotoxicity towards non-cancerous cells, even without the inclusion of a targeting moiety. The results demonstrated a potentially efficient anticancer drug delivery system with selectivity to breast cancer cells.

## 4. Challenges and Perspectives

The use of nanomedicine in cancer therapy sounds promising and the integration of Au and AgNPs within a chitosan matrix holds significant potential for cancer therapy advancements. Certain obstacles must be conquered in order to fully utilize their capabilities. In order to solve the difficulties that arise from the integration of medicinal and imaging elements in a nanocomposite, scientists need to develop innovative approaches to guarantee smooth operation between the different sections.

Based on the present overview, the therapeutic efficacy of the CS nanocomposite is significantly influenced by its stability and the sustained drug release characteristics. The continuous resolution of hard issues, such as ensuring controlled release kinetics, minimizing premature drug leakage, and sustaining drug concentrations at the target area, is essential. The majority of studies focus on encapsulating well-known anticancer drugs such as doxorubicin, 5-Fluorouracil, curcumin, and cisplatin, often neglecting other crucial medications. Therefore, it is prudent to consider a broader spectrum of anticancer drugs, including Daunorubicin for Kaposi’s sarcoma and myeloid leukemia [113], Vinorelbine for breast cancer and solid tumors [114], Mitoxantrone (for ovarian, breast, stomach, and leukemia) [115], Methotrexate for lung cancer [116], and Disulfiram for breast cancer [117]. Additionally, the most commonly used chitosan derivatives as a matrix phase in the nanocomposite carrier are carboxymethyl-forming CS and trimethyl CS for enhanced solubility, pH control, and loading capacity, while the antimicrobial activity is dependent only on the incorporated AgNPs. So, it is recommended to develop new CS derivatives with multi-bio characteristics including self-antimicrobial activities [118].

Moreover, both metals (such as AgNPs) and metal oxides (such as ZnO NPs) are mostly involved in the modifications of dispersed AuNPs along the CS nanocomposites for improving the entrapment efficiency of the anticancer drugs. Hence, it is necessary to find alternative materials with high performance. There are several effectual carbon-based materials such as graphene oxide derivatives and carbon nanotubes in addition to quantum dots proving their biological properties, which can be used for the modification of AuNPs. These materials are expected to offer protection for the entrapped drug molecules from degradation, as well as enhance the solubility, biocompatibility, photostability, and photothermal properties [119]. It is recommended that scientists collaborate across various fields for continuous advancements in nanotechnology to effectively harness the medicinal potential of these innovative nanocomposites.

Finally, one further challenge associated with the transformation of this technology to the commercial market is the absence of regulatory guidelines and standards for ensuring the safe production and quality control of these treatments. Consequently, it is obligatory to establish legal protocols for the development and clinical application of chitosan nanocomposites.

## 5. Conclusions

Nanomedicine products are potentially lucrative prospects for attaining advanced targeting methods and multifunctionality. The chitosan matrix phase can be modified to enhance the composite targeting properties, making it more sensitive to pH. The nanocomposite surface has been actively functionalized to selectively target cancer cells, allowing for precise contact with cancer cell receptors. Furthermore, the surface of the nanocomposite underwent active modification, facilitating its selective interaction with cancer cell receptors, thereby permitting targeted delivery to cancer cells. The incorporation of gold and silver nanoparticles within the chitosan nanocomposite holds considerable promise for anticancer drug delivery. The present review briefly outlines the ongoing efforts in the field of nanomedicine by enabling the transformation of nanocomposites into practical applications for personalized cancer treatment.

## Figures and Tables

**Figure 1 molecules-29-02393-f001:**
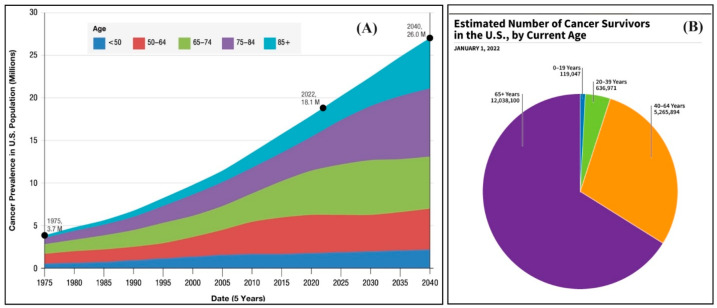
(**A**) Cancer prevalence and projections in U.S. from 1975 to 2040; (**B**) estimated number of 2022 cancer survivors by age.

**Figure 2 molecules-29-02393-f002:**
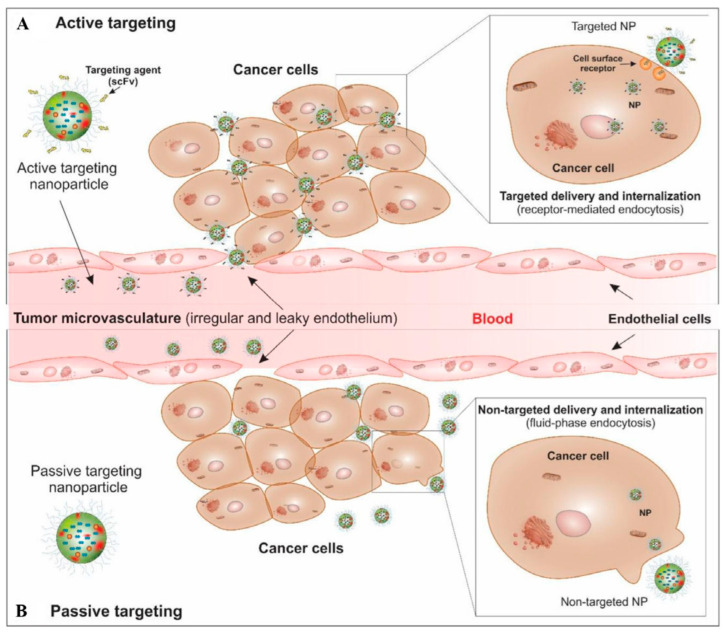
A schematic diagram for the targeted therapy of solid tumors by passive and active targeting mechanisms. Reproduced with permission [56], Copyright 2024, Elsevier.

**Figure 3 molecules-29-02393-f003:**
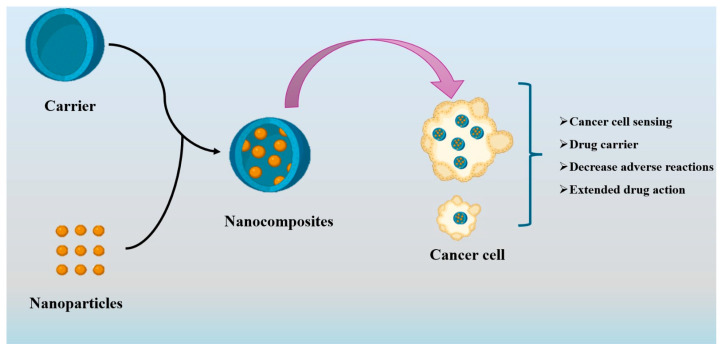
Proposed diagram for the synthesis of the nanocomposite carrier in cancer treatment.

**Figure 4 molecules-29-02393-f004:**
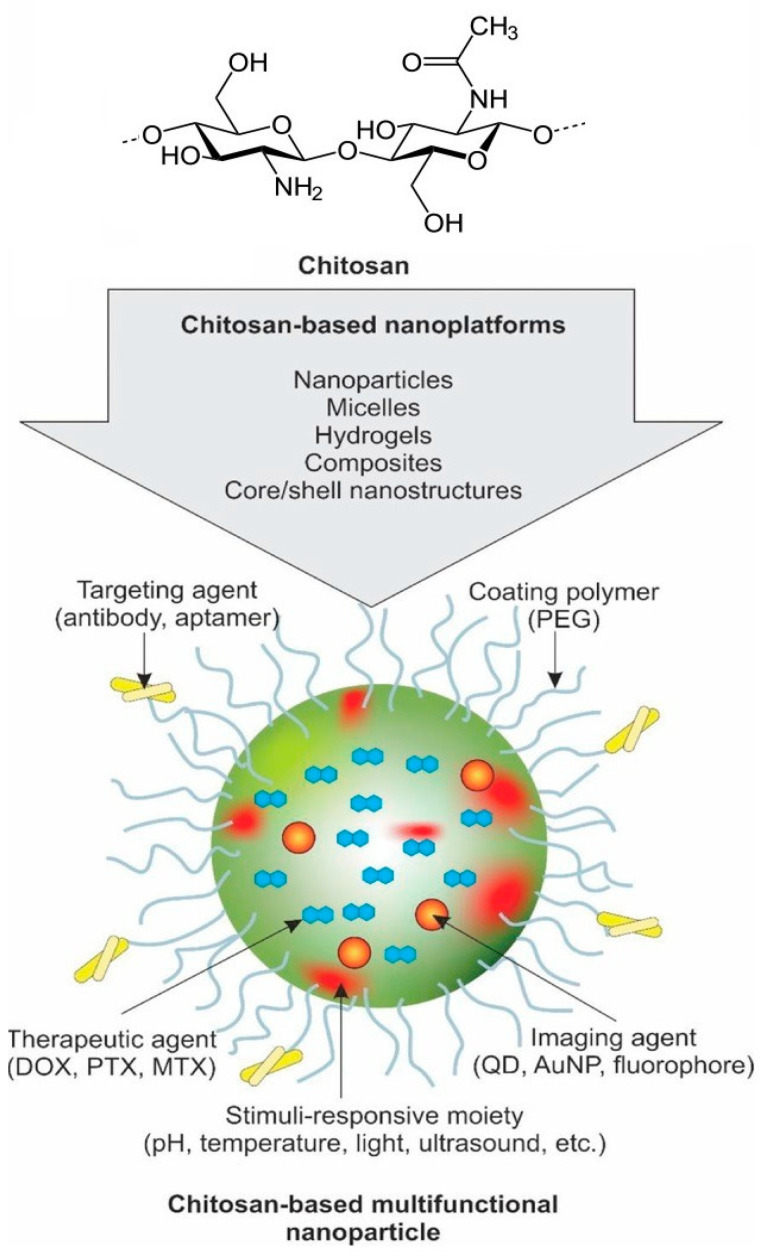
A schematic representation of CS-based multifunctional targeted nanosystem for simultaneous imaging and therapy of cancer. Reproduced with permission [56], Copyright 2024, Elsevier.

**Figure 5 molecules-29-02393-f005:**
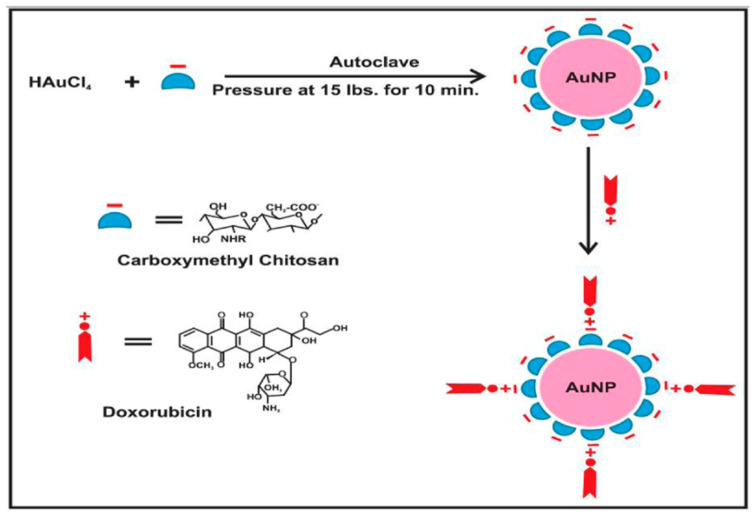
Schematic diagram showing gold nanoparticles (AuNPs) stabilized in CM-chitosan and subsequent loading of cationic doxorubicin on AuNPs (the negative sign “–” refers to anionic carboxylic groups in carboxymethyl chitosan and the positive sign “+” refers to cationic NH_2_ groups in doxorubicin) [84].

**Figure 6 molecules-29-02393-f006:**
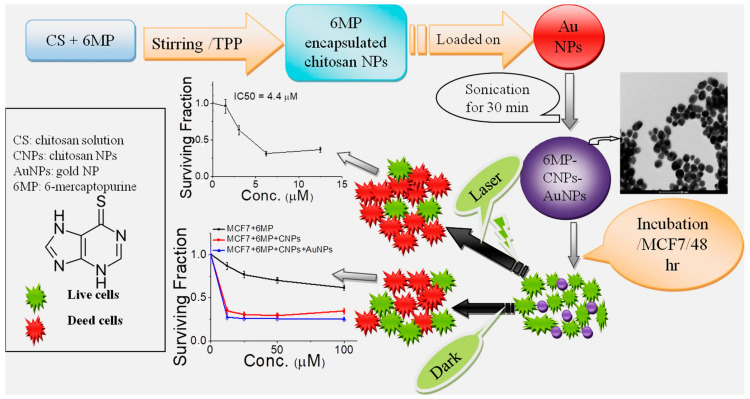
Illustration of preparation of chitosan-encapsulated 6MP with subsequent loading on AuNPs to form 6MP-CNPs-AuNPs nanocomposites, as well as treatment of MCF7 with the as-prepared nanocomposites in the absence and presence of DPSS laser for chemo-photothermal therapy [88].

**Figure 7 molecules-29-02393-f007:**
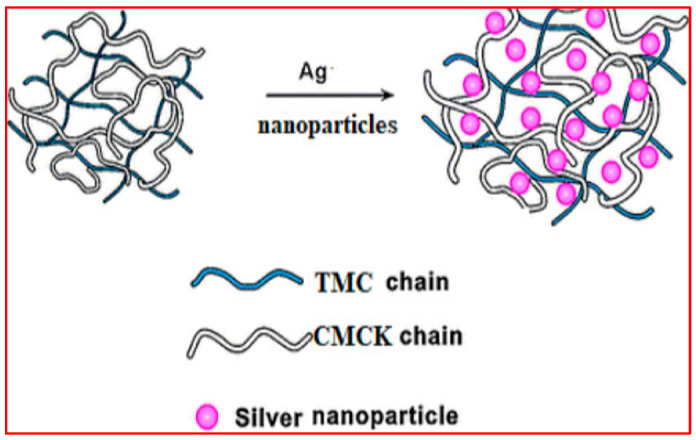
Schematic diagram showing the formation of silver nanocomposites (SNC) through the incorporation of Ag nanoparticles in (TMC-CMCK) PEC. Reproduced with permission [99]. Copyright 2024, Elsevier.

**Table 1 molecules-29-02393-t001:** Chitosan AuNP composite modifications and advantages for anticancer drug delivery.

Chitosan Matrix Phase Modifications	AuNPs Dispersed Phase Modifications	Loaded Anticancer Drug	Advantages of Modification	Ref.
Modified carboxymethyl chitosan CM-CS	AuNPs	Dox.	- More acidic pH sensitive for targeting Dox to cancer cells. - enhances cytotoxicity by increasing interaction with Dox. - reducing and capping agent for AuNPs - Enhancing cancer cell absorption of Dox.	[84]
CS	AuNPs + Ag NPs (bimetallic)	5-Fluroacil	- Bimetallic alloy nanoparticles possess high catalytic and electronic properties. - Enhancing AuNPs’ photothermal properties (PTT). - They prolong 5-FU release to 67.6% after 72 h. - Bimetallic alloys are more effective in cytotoxicity. - These nanomaterials enhance antimicrobial properties. - Could be used in chemical sensors due to their similar lattice constants.	[85]
CS	AuNPs + zinc oxide (ZnO)	Modified curcumin, Hydrazino curcumin derivative (HCUR)	- ZnO incorporation and curcumin modification improve drug entrapment efficiency (EE %) - prolong release and fasting release in acidic mediums. - Enhance cytotoxicity against HCT-116 cells (colon carcinoma) and HepG-2 cells (hepatocellular cancer).	[86]
COS chitosan oligosaccharide	AuNPs	Paclitaxel (PTX).	- AuNPs green synthesis is environmentally friendly, - cost-effective with zero energy based and less consuming time. - Enhances photothermal properties. - High solubility in neutral conditions with small weight compared to poor soluble CS. - Unique bioactivities, including antimicrobial, antitumor, and antiviral activities. - COS can couple with polymer subunits and its cationic nature allows ionic crosslinking. - COS is enhanced with hydrophobic residues in the cells for self-assembled nanoparticles targeting tumors. - COS marine biopolymer sources are used for gold nanoparticle green synthesis, ensuring high biocompatibility.	[87]
CS modified to chitosan nanoparticles (CNPs)	AuNPs	6-mercaptopurine (6MP)	- CNPs form complexes with 6MP, reducing side effects. - facilitating chemo-photothermal therapeutic effects - Improving efficacy compared to individual treatments. - AuNPs enhance cytotoxicity against cancer cells.	[88]

**Table 2 molecules-29-02393-t002:** Chitosan-AgNP bio-nanocomposite modifications and advantages for anticancer drug delivery.

Matrix Phase Modification	Dispersed Phase Modification	Encapsulated Anticancer Drug	Advantages of Modification	Ref.
N, N, N trimethyl chitosan modified (TMC) + carrageen	AgNPs	5-FU	- Chitosan modified with N, N, N trimethyl results in pH-sensitive control of intestinal discharge and targeting 5-FU. - increase drug loading capacity and drug release amount but less prolonged. - Green biosynthesis based AgNPs reduce toxic agents.	[99]
CS	CNT + AgNPs	5-FU	- CNT relatively strong binding 5-FU resulting in more prolonged release, more effective cytotoxic effect and antimicrobial compared to AgNPs without them. - CNT protects entrapped drug molecules against denaturation or degradation over the time duration of the delivery process. - CNT has large inner pore volumes loading more than one therapeutic agent at the same time. - Possess unique electronic, mechanical, and structural properties to ideal nanomaterials for drug delivery carriers.	[100]
CS	GO + AgNPs	5-FU	- GO’s excellent mechanical properties with surface area containing oxygen enable drug carrier applications. - GO is a promising photothermal agent due to high absorption cross-sections in the near-infrared (NIR) region. - GO-introduced RT-APM process rapidly synthesizes CS/AgNPs without hazardous chemicals.	[101]
CS	GO + AgNPs	Dox	- AgNPs dispersion enhances drug penetration and loading capacity due to creating high free spaces and pores. - AgNPs nanocomposite beads provide prolonged DOX release due to electrostatic attraction, and hydrogen bonding with DOX. - The presence of AgNPs in GO surface sheets enhances antibacterial properties and cytotoxicity against cancer cells.	[102]
N, N, N trimethyl chitosan modified (TMC) + carrageen	AgNPs	Curcumin	- Release in intestinal pH 7.4 sensitivity higher than gastric pH. - TMC amino groups ionized in acidic mediums, carrageenan sulfate exhibits anionic interaction, resulting in high crosslinking density and reducing water absorption. - TMC exhibits higher water solubility due to quaternization and van der Waals interactions. - TMC methyl groups positively interact with hydrophobic negative charged CUR, enhancing drug encapsulation and loading capacity. - High drug interaction intensifies hydrophobic curcumin-chitosan hydrogen bonding, improving solubility.	[103]
CS	AgNPs	Cisplatin (CIS)	- Limited, less sensitive CIS conjugation to be released at low pH targets breast cancer cells, causing 80% cell death with minimal cytotoxicity. - Chitosan functionalized AgNPs exhibit small size, high positive zeta potential, and encapsulate more than 80% cisplatin.	[104]

## Data Availability

Not applicable.

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
