# Peer review of "Recent Advancements in Metallic Au- and Ag-Based Chitosan Nanocomposite Derivatives for Enhanced Anticancer Drug Delivery"

_molecules, 2024, doi:10.3390/molecules29102393_

Round 1

Reviewer 1 Report

Comments and Suggestions for Authors

The manuscript by Elmeligy M.A. et al., entitled: Recent progress in chitosan-metallic Au and Ag nanocomposites for efficient delivery of anticancer drugs, presents an interesting and useful review on the anticancer potential of chitosan-metallic Au and Ag nanocomposites.

The manuscript is well-written and documented, offering the most recent insights in terms of anticancer nanotechnology. The references are sufficient and eloquent for the addressed theme.

The English language is very accurate, but the following sentence (lines 165-167) is unclear: Conventional approaches to cancer treatment are associated with negative consequences due to the high concentrations of medications in specific anatomical places, whereas diagnostic techniques are both expensive and time-intensive [6]. Please, explain what you mean by specific anatomical places.

The authors present the discussed nanocomposite as very efficient systems for anticancer drugs, but still, there is a huge gap between scientifically exposed data and the drugs industry that is not able to provide these products on the market. The authors mentioned the disadvantages of these forms for mass production and also tried to offer a solution, but they are still pale in comparison to the advantages mentioned in the manuscript.

At least, regarding lines 162-164, the authors should be more specific: There are still difficulties in the fabrication of nanoparticles (NPs) for purposes such as imaging, drug delivery, diagnosis, and treatment. The development of nanoparticles remains a challenging effort due to the observed instability of several nanoformulations in biological fluids [30, 32].

Also, the Challenges and perspectives part is evasive in comparison to the well-informed rest of the manuscript. The manuscript should indeed be about recent progress, as the title suggests, but considering the lack of availability for these kinds of products, the limitations should be exposed more specifically.

Author Response

1- The manuscript by Elmeligy M.A. et al., entitled: Recent progress in chitosan-metallic Au and Ag nanocomposites for efficient delivery of anticancer drugs, presents an interesting and useful review on the anticancer potential of chitosan-metallic Au and Ag nanocomposites.

The manuscript is well-written and documented, offering the most recent insights in terms of anticancer nanotechnology. The references are sufficient and eloquent for the addressed theme.

The English language is very accurate, but the following sentence (lines 165-167) is unclear: Conventional approaches to cancer treatment are associated with negative consequences due to the high concentrations of medications in specific anatomical places, whereas diagnostic techniques are both expensive and time-intensive [6]. Please, explain what you mean by specific anatomical places.

Response: At first, special thanks to the reviewer for his positive response regarding our manuscript, and also for his professional comments.

Then about the anatomical places, it means that high doses of conventional medications cause side effects in different body tissues, such as the skin, hair, RBCs, bone, genitourinary system, and lymphatic system; all of these examples refer to specific anatomical places.

We have rewritten this sentence with more clarification as follow: (Page 5: Lines 194-199]

“Conventional approaches of cancer treatment are associated with negative consequences, including limited solubility, insufficient bioavailability, frequent deterioration, lack of specificity, and a drug resistance. In addition, the high doses of conventional medications cause adverse effects in in specific anatomical places, such as skin, hair, RBCs, bone, genitourinary system, and lymphatic system [6].”

2- The authors present the discussed nanocomposite as very efficient systems for anticancer drugs, but still, there is a huge gap between scientifically exposed data and the drugs industry that is not able to provide these products on the market. The authors mentioned the disadvantages of these forms for mass production and also tried to offer a solution, but they are still pale in comparison to the advantages mentioned in the manuscript.

At least, regarding lines 162-164, the authors should be more specific: There are still difficulties in the fabrication of nanoparticles (NPs) for purposes such as imaging, drug delivery, diagnosis, and treatment. The development of nanoparticles remains a challenging effort due to the observed instability of several nanoformulations in biological fluids [30, 32].

Response: We have modified this section (Page 4: lines 188-194) to be clearer, the nanoparticles specified for metal nanoparticles with adding two additional lines to specify the reasons which discussed before and supported by relevant references.

“There are still difficulties in the fabrication of metal nanoparticles (NPs) for purposes such as imaging, drug delivery, diagnosis, and treatment. The development of nanoparticles remains a challenging effort due to the observed instability of several nanoformulations in biological fluids [51, 53]. Biological fluids' high ionic strengths often induce NP aggregation and losing their colloidal stability, negatively affecting their function. The high content of biomacromolecules, including lipids, sugars, nucleic acids, and proteins, also affects NPs stability and viability for various applications [54]. Conventional approaches of cancer treatment are associated with negative consequences, including limited solubility, insufficient bioavailability, frequent deterioration, lack of specificity, and a drug resistance. In addition, the high doses of conventional medications cause adverse effects in in specific anatomical places, such as skin, hair, RBCs, bone, genitourinary system, and lymphatic system [6].”

3- Also, the Challenges and perspectives part is evasive in comparison to the well-informed rest of the manuscript. The manuscript should indeed be about recent progress, as the title suggests, but considering the lack of availability for these kinds of products, the limitations should be exposed more specifically.

Response: challenges and prospective part was rewritten for more clarification, with adding new references to be as the following:

Although the use of nanomedicine in cancer therapy sounds promising and exciting, the integration of Au and Ag NPs within a chitosan matrix holds significant potential for cancer therapy advancement. Certain obstacles must be conquered in order to fully utilize their capabilities. In order to solve the difficulties that arise from the integration of medicinal and imaging elements in a nanocomposite, scientists need to develop innovative approaches to guarantee smooth operation between the different sections.

Based on the present overview, the therapeutic efficacy of the CS nanocomposite is significantly influenced by its stability and the sustained drug release characteristics. The continuous resolution of hard issues such as ensuring controlled release kinetics, minimizing premature drug leakage, and sustaining drug concentrations at the target area is essential. The majority of studies focus on encapsulating well-known anticancer drugs such as Doxorubicin, 5-Fluorouracil, Curcumin, and Cisplatin, often neglecting other crucial medications. Therefore, it is prudent to consider a broader spectrum of anticancer drugs, including Daunorubicin for Kaposi's sarcoma and myeloid leukemia [113], Vinorelbine for breast cancer and solid tumors [114], Mitoxantrone (for ovarian, breast, stomach, and leukemia) [115], Methotrexate for lung cancer [116], and Disulfiram for breast cancer [117]. Additionally, the most common used chitosan derivatives as a matrix phase in the nanocomposite carrier are carboxymethyl-forming CS and trimethyl CS for enhanced solubility, pH control, and loading capacity, while antimicrobial activity de-pendent only on the incorporated Ag NPs. So, it’s recommended to develop new CS derivatives with multi-bio characteristics including self-antimicrobial activities. [118].

Moreover, both metals (such as Ag NPs) and metal oxides (such as ZnO NPs) are mostly involved for the modifications of dispersed Au NPs along the CS nanocomposites for improving the entrapment efficiency of the anticancer drugs. Hence, it is necessary to find alternative materials with high performance. There are several effectual carbon-based materials such as graphene oxide derivatives and carbon nanotube in addition to quantum dots proved their biological properties, which can be used for the modification of Au NPs. These materials are expected to offer a protection for the entrapped drug molecules from degradation, as well as enhance solubility, biocompatibility, photostability, and photothermal properties [119]. It is recommended for scientists to collaborate across various fields and continuous advancements in nanotechnology to effectively harness the medicinal potential of these innovative nanocomposites.

Finally, one further challenge associated with the transformation of this technology to the commercial market is the absence of regulatory guidelines and standards for ensuring the safe production and quality control of these treatments. Consequently, it is obligatory to establish legal protocols for the development and clinical application of the chitosan nanocomposites.

Thanks to the reviewer for his valuable comments and suggestions, we hope that the current version of the revised manuscript can be accepted.

Reviewer 2 Report

Comments and Suggestions for Authors

This review reports advances in chitosan-metallic nanocomposites as an anticancer drug delivery system. The manuscript includes valuable information for the public of this journal. However, some aspects need to be improved before being published. Here are the main observations:

 1.     The title is confusing and not consistent with the content of the paper. For example, the paper reviews chitosan derivatives used to manufacture nanoparticles by incorporating nanometals. However, the title only refers to chitosan, not to chitosan derivatives. I consider that the title can be improved to be more precise.

2.     The following suggestions should improve the writing coherence and of the introduction:

-        The document requires consistency in the structure of the wording. For example, the sentence "The oral delivery route is a straightforward and preferable method for treating gastrointestinal tumors (page 2, lines 82-83)" seems to be "loose" as it had not discussed specific tumors up to that point (2) the sentence "The primary novel methodologies now rely on the utilization of nanomedicines, and nanocarriers ought to be expanded for the transportation of biomolecules (page 3, lines 100-103)" abruptly indicates the use of nanomedicine as an alternative methodology without first discussing this topic in the introduction.

-        Page 3, line 118: "... are increasingly being utilized in clinical applications as distinct carriers". It is recommended that the text be concise.

-        Page 3, lines 102-103 ("The utilization of polymer-prodrug systems…"). The authors briefly mention polymeric systems but then return to this topic until line 130 (page 4).

3.     Page 3, lines 130-131. The authors mention, "Therefore, it is widely acknowledged that natural polysaccharides and natural biopolymers are advantageous stabilizers." Could you please explain the reasons for this assertion and include some references to support this?

4.     Page 6-7, lines 246-252. This sentence is very long, which causes confusion and makes it difficult to understand the information.

5.     Figure 4. The chitosan molecule is written in its fully deacetylated version; however, on lines 245-246, chitosan is referred to as a copolymer containing two types of units. I suggest that the molecular structure of chitosan be correctly placed, where the deacetylated units are also shown in Figure 4.

6.     Page 7. Lines 255-257. The authors mention, "Chitosan exhibits considerable potential as a viable option for the formation of nanoparticles, enhancing the stability and bioavailability of bioactive compounds." Could you please specify the characteristics of chitosan to make it a viable option for forming nanoparticles? What kind of methods exist for the formation of chitosan nanoparticles? Although this is not a review of nanoparticle preparation methods, I consider it necessary to add an epigraph where the main methods for the preparation of nanoparticles from this biopolymer are briefly mentioned. A reference that may help you is the following: https://www.mdpi.com/2073-4360/10/3/235

Comments on the Quality of English Language

The document has several grammar and writing errors; please review it carefully before submitting it again. For example:

page 3, line 118: ... are increasingly being utilized in clinical applications as distinct carriers.

Author Response

This review reports advances in chitosan-metallic nanocomposites as an anticancer drug delivery system. The manuscript includes valuable information for the public of this journal. However, some aspects need to be improved before being published. Here are the main observations:

  1. The title is confusing and not consistent with the content of the paper. For example, the paper reviews chitosan derivatives used to manufacture nanoparticles by incorporating nanometals. However, the title only refers to chitosan, not to chitosan derivatives. I consider that the title can be improved to be more precise.

Response: Thanks to the reviewer for these worthy comments. For the title, we have improved it to be more precise.

“Recent advancements in metallic Au and Ag -based chitosan nanocomposite derivatives for enhanced anticancer drug delivery”

  1. The following suggestions should improve the writing coherence and of the introduction:

-        The document requires consistency in the structure of the wording. For example, the sentence "The oral delivery route is a straightforward and preferable method for treating gastrointestinal tumors (page 2, lines 82-83)" seems to be "loose" as it had not discussed specific tumors up to that point (2) the sentence "The primary novel methodologies now rely on the utilization of nanomedicines, and nanocarriers ought to be expanded for the transportation of biomolecules (page 3, lines 100-103)" abruptly indicates the use of nanomedicine as an alternative methodology without first discussing this topic in the introduction.

Response: We have improved this section with mention types of gastrointestinal tumors, and also we have discussed the topic of nanomedicine and nanocarriers in the introduction section with updated references as follow:

Pages 2, 3: Lines 83-92

“The oral delivery route is a straightforward and preferable method for treating various gastrointestinal tumors such as colorectal cancer, stomach cancer, pancreatic cancer, and anal cancer [10]. Each type of cancers may require different treatment approaches depending on its location, stage, and other factors [11]. Therefore, the drugs can be absorbed directly through the gastrointestinal tract, where many of these tumors are located, potentially leading to higher drug concentrations at the tumor site. This targeted delivery approach enhances the therapeutic effect while minimizing systemic toxicity [12]. However, the oral administration of anticancer medications presents several obstacles, including the need to improve bioavailability (solubility and/or permeability), reduce enzymatic degradation, and achieve targeted delivery to specific sites within the gastrointestinal tract [13, 14].”

Page 3: Lines 105-126

“Nanomedicines and nanocarriers represent encouraging approaches for delivering anticancer medications with improved accuracy and effectiveness [18, 19]. Nanomedicines involve the encapsulation or attachment of therapeutic substances to nanoparticles, while nanocarriers are specialized vehicles crafted to convey drugs to precise locations within the body [20, 21]. In the realm of anticancer treatment delivery, both nanomedicines and nanocarriers present multiple benefits [22]. Their nanoscale size enables them to traverse biological barriers more efficiently, including tumor tissues with impaired blood vessel function. This enhanced permeability and retention (EPR) effect enables preferential accumulation of the drug at the tumor site, minimizing exposure to healthy tissues and reducing systemic toxicity [23]. Furthermore, nanocarriers have the capability to be tailored to target specific molecular markers that are excessively expressed on cancer cells, thereby refining the precision of drug delivery. Moreover, nanocarriers can protect encapsulated drugs from degradation and premature clearance in the bloodstream, prolonging circulation time and enhancing bioavailability. This feature of controlled release enables sustained delivery of drugs, maintaining therapeutic levels over an extended duration, and potentially decreasing the frequency of administration [24]. Additionally, the adaptability of nanomedicines and nanocarriers permits the simultaneous delivery of multiple drugs or therapeutic agents with complementary modes of action. This coordinated approach can bolster treatment effectiveness, surmount drug resistance, and diminish the likelihood of tumor recurrence [25]. The primary novel methodologies now rely on the utilization of nanomedicines and nanocarriers ought to be expanded for the transportation of biomolecules [26, 27].

- Page 3, line 118: "... are increasingly being utilized in clinical applications as distinct carriers". It is recommended that the text be concise.

Response: We have rewritten the sentence to be more concise as follow:

“Metallic nanoparticles, including gold nanoparticles (AuNPs) and silver nanoparticles (AgNPs), are utilized in clinical applications. This is due to their unique forms, sizes, and surface-dependent properties”

-  Page 3, lines 102-103 ("The utilization of polymer-prodrug systems…"). The authors briefly mention polymeric systems but then return to this topic until line 130 (page 4).

 Response: We have rearranged this section to be clearer as follow: [Page 3: Lines 127-126]

“The utilization of nanotechnology in anticancer drug delivery presents several challenges, including the encapsulation of hydrophilic or hydrophobic pharmaceuticals, regulation of drug release, protecting the loaded drugs from degradation, and improving drug absorption and penetration [28]. Nevertheless, there are certain limitations for using nanotechnology including poor drug loading, rapid or burst release before reaching the intended target, and probability of clearance through renal filtration owing to nanoscale dimensions. The utilization of polymer-prodrug systems and the integrating of polymer nanocomposites can overcome these drawbacks and enables the creation of a depot beneath the skin, facilitating the controlled and gradual release of drugs with minimal adverse effects on the surrounding region [29, 30].

  1. Page 3, lines 130-131. The authors mention, "Therefore, it is widely acknowledged that natural polysaccharides and natural biopolymers are advantageous stabilizers." Could you please explain the reasons for this assertion and include some references to support this?

 Response: We have improved this part by adding explanation for the use of natural biopolymers as stabilizers for metallic nanoparticles and supported by updated references as follows: Page 4: Lines 150–157

“Hence, it is necessary to incorporate further organic or biological surface coatings to enhance the stability of AuNPs and AgNPs [36, 37]. Thus, it is widely acknowledged that natural polysaccharides and natural biopolymers are advantageous stabilizers. Therefore, as metal nanoparticles tend to be aggregated in biological fluids, and this can be overcome by the generated electrostatic attractive forces between polysaccharide (such as amino groups of chitosan) and metallic nanoparticle [38]. Consequently, this can provide an effective driving force for the formation and stabilization of the Au and Ag NPs [39, 40]."

  1. Page 6-7, lines 246-252. This sentence is very long, which causes confusion and makes it difficult to understand the information.

Response: We have rewritten the sentence to be clearer as follow: Page 7: Lines 269–272

“Research findings indicate that nanocomposites possess the ability to breach biological barriers and deliver drugs to specific cells within the tumor microenvironment. This consequently enhances the effectiveness of cancer therapy [71]”.

  1. Figure 4. The chitosan molecule is written in its fully deacetylated version; however, on lines 245-246, chitosan is referred to as a copolymer containing two types of units. I suggest that the molecular structure of chitosan be correctly placed, where the deacetylated units are also shown in Figure 4.

Response: We have changed the molecular structure of chitosan as a copolymer containing two types of units.

  1. Page 7. Lines 258-260. The authors mention, "Chitosan exhibits considerable potential as a viable option for the formation of nanoparticles, enhancing the stability and bioavailability of bioactive compounds." Could you please specify the characteristics of chitosan to make it a viable option for forming nanoparticles? What kind of methods exist for the formation of chitosan nanoparticles? Although this is not a review of nanoparticle preparation methods, I consider it necessary to add an epigraph where the main methods for the preparation of nanoparticles from this biopolymer are briefly mentioned. A reference that may help you is the following: https://www.mdpi.com/2073-4360/10/3/235

 Response: We have improved this part by adding the characteristics of chitosan that make it a viable option for forming nanoparticles. Also, we have mentioned the used methods for the formation of CS with supporting references as follow (Page 7: Lines 281-289):

“Chitosan has drawn much attention in diverse medical and pharmaceutical fields such as tissue engineering, wound healing, and drug delivery [74]. This is due to its excellent characteristics including low-cost production, non-toxic, mucoadhesive, biocompatibility, biodegradability, low immunogenicity, and antimicrobial and anticancer activities [41, 46]. Owing to its cationic nature, adequate stability, versatility in formulation and ease of modification, CS exhibits considerable potential as a viable option for the formation of nanoparticles. Various methods have been employed to fabricate CS NPs, such as ionotropic gelation, spray drying, cross-linking via water-in-oil emulsion, reverse micelle formation, emulsion-droplet coalescence, nanoprecipitation, and self-assembling techniques [75, 76].”

Thanks to the reviewer for his valuable comments and suggestions, we hope that the current version of the revised manuscript can be accepted.